# *SMAD3* Hypomethylation as a Biomarker for Early Prediction of Colorectal Cancer

**DOI:** 10.3390/ijms21197395

**Published:** 2020-10-07

**Authors:** Muhamad Ansar, Chun-Jung Wang, Yu-Han Wang, Tsung-Hua Shen, Chin-Sheng Hung, Shih-Ching Chang, Ruo-Kai Lin

**Affiliations:** 1Ph.D. Program in the Clinical Drug Development of Herbal Medicine, Taipei Medical University, Taipei 110301, Taiwan; muhamadanshar919@gmail.com; 2School of Pharmacy, College of Pharmacy, Taipei Medical University, Taipei 110301, Taiwan; aquaholic0322@gmail.com (C.-J.W.); tsunghuashen19960831@gmail.com (T.-H.S.); 3School of Medicine, Tzu Chi University, Hualien City 970374, Taiwan; g7899nek@gmail.com; 4Department of Surgery, School of Medicine, College of Medicine, Taipei Medical University, Taipei 110301, Taiwan; hungcs@tmu.edu.tw; 5Division of General Surgery, Department of Surgery, Shuang Ho Hospital, Taipei Medical University, New Taipei City 235041, Taiwan; 6Division of Colon and Rectal Surgery, Department of Surgery, Taipei Veterans General Hospital, Taipei 112201, Taiwan; 7Ph.D Program in Drug Discovery and Development Industry, College of Pharmacy, Taipei Medical University, Taipei 110301, Taiwan; 8Master Program for Clinical Pharmacogenomics and Pharmacoproteomics, Taipei Medical University, Taipei 110301, Taiwan; 9Clinical Trial Center, Taipei Medical University Hospital, Taipei 110301, Taiwan

**Keywords:** Mothers Against Decapentaplegic Homolog 3 *(SMAD3*), DNA methylation, hypomethylation, biomarker, circulating cell-free DNA (ccfDNA), colorectal cancer (CRC), early detection

## Abstract

The incidence and mortality rates of colorectal cancer (CRC) have been high in recent years. Prevention and early detection are crucial for decreasing the death rate. Therefore, this study aims to characterize the alteration patterns of mothers against decapentaplegic homolog 3 (*SMAD3*) in patients with CRC and its applications in early detection by using a genome-wide methylation array to identify an aberrant hypomethylation site in the intron position of the *SMAD3* gene. Quantitative methylation-specific polymerase chain reaction showed that hypomethylated *SMAD3* occurred in 91.4% (501/548) of Taiwanese CRC tissues and 66.6% of benign tubular adenoma polyps. In addition, *SMAD3* hypomethylation was observed in 94.7% of patients with CRC from The Cancer Genome Atlas dataset. A decrease in circulating cell-free methylation *SMAD3* was detected in 70% of CRC patients but in only 20% of healthy individuals. *SMAD3* mRNA expression was low in 42.9% of Taiwanese CRC tumor tissues but high in 29.4% of tumors compared with paired adjacent normal tissues. Hypomethylated *SMAD3* was found in cancers of the digestive system, such as liver cancer, gastric cancer, and colorectal cancer, but not in breast cancer, endometrial cancer, and lung cancer. In conclusion, *SMAD3* hypomethylation is a potential diagnostic marker for CRC in Western and Asian populations.

## 1. Introduction

Colorectal cancer (CRC) has become an increasing global health burden in recent years. In 2018, it ranked third in terms of incidence and second in terms of death rate. The mortality rate of CRC is the third-highest for cancers in the United States and Taiwan [1,2]. Annually, more than 1.8 million new cases are diagnosed and 881,000 deaths are recorded worldwide [3]. Furthermore, the rate of increase in new cases and deaths is estimated to grow to 60% by 2030 [4]. Therefore, prevention and early detection are crucial. In the early stages, a cure is often possible [5]. Screening programs are viewed as a way to decrease CRC mortality [6]. According to a study, 70% of patients with CRC develop the disease sporadically, but 30% of patients develop it through genetic susceptibility and heredity. Abnormal alterations of genetics and epigenetics, such as chromosomal instability, microsatellite instability, CpG island methylator phenotype, and DNA methylation, play a significant role in colorectal tumorigenesis [7]. Among epigenetic changes, DNA methylation of the promoter zone is considered the first occurrence. Because of stability and a specific shift in DNA methylation, it has emerged as a potential biomarker for CRC [8,9]. Combining circulating cell-free DNA (ccfDNA) with carcinoembryonic antigen is persuasive in diagnosis nowadays [10]. The U.S. Food and Drug Administration proved that *Septin 9* (*SEPT9*) serves as a putative biomarker in the early detection of CRC, with significant sensitivity (71.1–95.6%) and specificity (81.5–99%) [11]. As described previously, 86.1% of the CRC patients in Taiwan showed BEN Domain Containing 5 (*BEND5*) hypermethylation [12]. Thus, sensitive biomarkers are valuable in Asian groups. Methylated ccfDNA is beneficial not only for early diagnosis but also for prognosis in metastatic CRC [13]. These novel findings motivate us to conduct an in-depth study to identify new potential biomarkers for detecting CRC early. Thus, in this study, we used the human methylation 450K array to classify the alteration patterns of the protein-coding gene in CRC: mothers against decapentaplegic homolog 3 (*SMAD3*). The *SMAD3* methylation level in CRC tumors, which was half that of adjacent normal colorectal tissue, was defined as hypomethylation.

*SMAD3* is related to the transforming growth factor-β (TGF-β) signaling pathway, which is connected to tumor development [14]. Moreover, *SMAD3* can promote cancer progression in non-small-cell lung cancer through regulation of paired box 6 [15]. In this study, DNA methylation decreased in Western patients from The Cancer Genome Atlas (TCGA) and Taiwanese patients with CRC. The role of *SMAD3* hypomethylation in CRC is unclear. No study has focused on the relationship between the *SMAD3* methylation level and CRC diagnosis. Thus, determining the methylation level, RNA expression level, and clinical data correlation is the primary purpose of this study.

## 2. Results

### 2.1. SMAD3 Was Identified from Taiwanese and Western Patients with CRC through Genome-Wide Methylation Analysis

We set three criteria to classify potential CRC genes: (1) hypomethylation in Taiwanese patients with CRC, (2) a methylation level close to 0.2 in CRC tissues, and (3) hypomethylation in Western patients with CRC (Figure 1). First, we used the Illumina Infinium HumanMethylation450 BeadChip array to verify target genes from 26 pairs of cancerous and noncancerous tissues. In total, 626 genes were hypomethylated when the ΔAvg_β (Tumor–Normal) was less than −0.25. Among these genes, eight (*SMAD3*; Acyl-CoA Thioesterase 7, *ACOT7*; RAS P21 Protein Activator 3, *RASA3*; UDP-GlcNAc:BetaGal Beta-1,3-N-Acetylglucosaminyltransferase Like 1, *B3GNTL1*; Phosphatidylinositol Glycan Anchor Biosynthesis Class B, *PIGB*; Mitogen-Activated Protein Kinase Kinase Kinase 5, *MAP3K5*; Lipin 1, *LPIN1*, and Myosin Binding Protein C3, *MYBPC3*) were selected when the Avg_β value was high in noncancerous colorectal tissues (Avg_β of normal tissue was >0.5). Second, we used the same criteria to analyze TCGA Illumina Infinium HumanMethylation450 BeadChip array data from 38 pairs of Western cancerous and noncancerous tissues. When ΔAvg_β (Tumor–Normal) was less than −0.25, a total of 7105 genes were hypomethylated in the TCGA dataset. The outcome included the same eight genes that were selected from both Taiwanese and TCGA patients. Among these eight genes, *SMAD3* had the smallest Avg_β in tumor (0.12) and highest Avg_β in normal (0.52) tissues. According to the datasets from TCGA, *SMAD3* was hypomethylated in cancers of the digestive system, such as liver cancer, gastric cancer, colon cancer, and rectal cancer. However, the role of *SMAD3* methylation in cancer is still unclear. Consequently, additional studies on epigenetic changes and mRNA expression of *SMAD3* are required.

### 2.2. Methylation Level of SMAD3 in Tissues from Taiwanese Patients with CRC

*SMAD3* promotes cancer progression through the TGF-β signaling pathway [14]. Therefore, further investigations of DNA methylation alterations and mRNA expression were performed in both Taiwanese and Western patients with CRC. The Illumina Infinium HumanMethylation450 BeadChip array revealed only one methylation difference at cg24032190 based on a study of 26 Taiwanese paired cancerous and noncancerous tissues under the following criteria: Avg_β (Normal) > 0.5 and Avg_β (Tumor) < 0.2. This CpG site showed the values of Avg_β (Normal) and Avg_β (Tumor) to be 0.524 and 0.119, respectively. The CpG site is located at the gene body region +12535 (array probe 6, *p* = 0.036) of *SMAD3*. The represented CpG site (region +12535, array probe 6) in the Illumina methylation array was commercially selected. Not all CpG sites can be detected with an Illumina methylation array because of its limitation to 450,000 CpG methylation sites. However, a specific methylated primer design and sequencing revealed that several CpG sites near region +12535 showed decreased methylation in CRC tumor tissues. In total, 45 CpG sites were verified in *SMAD3*, but only cg24032190 showed a significant hypomethylation difference between cancerous and noncancerous tissues (*p*
≤ 0.001). The results were confirmed in the heatmap (Figure 2A).

The result of quantitative methylation-specific polymerase chain reaction (QMSP) assays in 548 patients with CRC and nine patients with benign tubular adenoma showed that the incidence rate of hypomethylation was 91.4% (501/548) in Taiwanese patients with CRC, and the methylation level of *SMAD3* in cancerous tissues was half that in noncancerous paired tissues (Figure 2B). Bar graphs of methylation levels for each Taiwanese CRC patient have been added to Appendix A. Hypomethylated *SMAD3* was observed in 66.6% (6/9) polyps obtained from Taipei Veterans General Hospital. Regarding the methylation trend (Figure 2B), the methylation level of *SMAD3* in polyps was between that of normal and tumor tissues. The results showed a significant difference when comparing normal tissue with tumors and polyps with tumors (*p* ≤ 0.001).

### 2.3. Methylation Level of SMAD3 in CRC Tissues from TCGA Datasets

To ensure the correlation of *SMAD3* hypomethylation in Western CRC patients, we used the Illumina Infinium HumanMethylation450 BeadChip array dataset to investigate the data of 38 colorectal paired tissues and 314 CRC tissues from TCGA. According to the results of the aforementioned analysis, we demonstrated the methylation levels using a heatmap. cg24032190, which is located on the gene body of *SMAD3*, showing that they were hypomethylated in 94.7% (36/38) of CRC paired tissues and 92.0% (289/314) of nonpaired colorectal tumor tissues (Figure 3A,B). Therefore, we inferred that *SMAD3* can be a predictive biomarker of CRC in Asian and Western populations.

### 2.4. Methylation Level of ccfDNA of SMAD3 in Plasma

We tested the methylation level of *SMAD3* ccfDNA in 200 μL of plasma extracted manually. The plasma from healthy participants without CRC was used as a healthy control and we compared the differences of methylated ccfDNA of *SMAD3* with CRC patients. A decrease in methylation was detected in 86.6% (13/15) of patients with CRC and 60% (9/15) of healthy participants (*p* = 0.041) (Figure 4A,B). The sensitivity was 86.6% (13/15) and the positive predictive value (PPV) was 59% (13/22). When 1 mL of plasma was used, decreased methylation was detected in seven out of 10 (70%) patients with CRC, but in only two out of 10 (20%) healthy controls (*p* = 0.038) (Figure 4C,D). The sensitivity and PPV of 1 mL of plasma were 78.5% (11/14) and 64.7% (11/17).

### 2.5. SMAD3 mRNA Expression in Taiwanese CRC Paired Tissues and the TCGA Dataset

We analyzed *SMAD3* mRNA expression in 119 paired CRC tissues, noncancerous tissues, and nine samples obtained from patients with benign tubular adenoma. In 42.9% (51/119) of tissue samples, *SMAD3* mRNA expression in Taiwanese CRC tissues was half that in noncancerous tissues. In 29.4% (35/119) of paired tissues, the expression was reduced by half in Taiwanese cancerous tissues compared to noncancerous tissues. No significant difference was found between normal tissue and polyps, or between tumor tissue and polyps, in terms of mRNA expression (*p* = 0.20 and 0.57, respectively; Figure 5A). Bar graphs of the expression level in each Taiwanese CRC patient have been added to Appendix A. Additionally, we analyzed the mRNA expression of *SMAD3* in 41 paired normal and tumor tissues in patients with CRC obtained from TCGA. Low expression in the tumor versus normal tissue was found in TCGA datasets (*p* ≤ 0.001) (Figure 5B). We found that the hypermethylation of five CpG sites located at the promoter regions caused low mRNA expression. Additionally, we found two CpG sites located in the gene body, namely cg07890839 (R = −0.350, *p* = 0.031) and cg03947447 (R = −0.440, *p* = 0.006), that also led to low expression. The results of the Pearson correlation test revealed a significant negative correlation between *SMAD3* mRNA expression and *SMAD3* hypermethylation in the cg18603446 promoter region −1133 (array probe 1, *p* = 0.031), cg01710852 promoter region −525 (array probe 2, *p* = 0.029), cg017119488 promoter region −413 (array probe 3, *p* = 0.004), cg013331691 promoter region −122 (array probe 4, *p* = 0.050), cg017092056 promoter region −94 (array probe 5, *p* = 0.005), cg07890839 gene body region +1331 (array probe 9, *p* = 0.031), and cg03947447 gene body region +11421 (array probe 10, *p* = 0.006). By contrast, the correlation was positive in the *SMAD3* hypomethylation cg24032190 gene body region +12535 region (array probe 6, *p* = 0.036) and cg25547520 gene body region +927 (array probe 7, *p* = 0.036) (Figure 6).

### 2.6. Clinical Characteristics of DNA Methylation and mRNA Expression

To investigate the relationship between DNA methylation and mRNA expression in terms of the clinical characteristics of Taiwanese patients with CRC and TCGA datasets, we used Pearson’s chi-squared test. The results revealed that *SMAD3* hypomethylation can be detected in both early and late stages of CRC (Table 1 and Appendix A). *SMAD3* hypomethylation is commonly observed in several clinical parameters, such as age, ethnicity, sex, tumor type, tumor stage, tumor size, regional lymph node metastasis, distant metastasis, differentiation grade, vascular invasion, location, microsatellite instability (MSI), and kirsten rat sarcoma viral oncogene homologue (KRAS) mutation. A comparison of the Taiwanese and TCGA datasets revealed similar results (Table 1 and Appendix A). An alteration in *SMAD3* mRNA can be observed in all stages of colorectal cancer. The results of methylation imply that *SMAD3* can play a vital role in detecting CRC. However, the technical difficulties in assessing *SMAD3* mRNA preclude its widespread application as a current potential biomarker.

### 2.7. Methylation Level of SMAD3 in Different Cancers

To check the methylation level of *SMAD3* in other cancers, we performed a DNA methylation analysis in esophageal, lung, endometrial, and breast cancer. According to the results, 62.5% (10/15) of patients with esophageal cancer had *SMAD3* hypomethylation. The trend is similar to that in CRC. However, the result was the opposite in endometrial cancer, where 60% (9/15) of patients showed *SMAD3* hypermethylation. Insignificant differences exist in lung (*p* = 0.37) and breast cancer (*p* = 0.16) between adjacent normal tissues and tumor tissues. Moreover, we analyzed TCGA data in other cancers. Hypomethylated *SMAD3* was found in cancers of the digestive system, such as liver cancer (8/12, 66.6%), gastric cancer (1/2, 50%), colon cancer (36/38, 94.7%), and rectal cancer (7/7, 100%) (Figure 7). In addition, the methylation levels in different cancers from TCGA datasets are listed in Table 2.

### 2.8. Correlation between Prognosis and SMAD3 Methylation Level

We investigated whether SMAD3 methylation is associated with survival in CRC. We stratified the overall survival of Taiwanese patients with CRC and patients in TCGA datasets into two subsets on the basis of SMAD3 methylation: low hypomethylation and no-hypomethylation groups. The results indicated that the survival rate of the no-hypomethylation group was higher than that of the hypomethylation group, especially among Taiwanese male, elderly, and late-stage CRC patients (Figure 8). However, these results were not found in Western populations.

## 3. Discussion

DNA methylation is suited to clinical application because of its stable characteristics [8]. Both hypermethylation and hypomethylation are independent processes critical to colorectal tumor formation [16]. Considering the prevalence and mortality rate of CRC, developing a novel biomarker is our main purpose. By performing Illumina Human Methylation 450K arrays on 26 paired Taiwanese CRC tissues, we found eight promising genes. In addition to *SMAD3*, the main gene in this paper, we collected information regarding seven other potential genes: *ACOT7*, *RASA3*, *B3GNTL1*, *PIGB*, *MAP3K5*, *LPIN1*, and *MYBPC3*. According to previous studies, a high expression of *ACOT7* is related to poor prognosis in acute myeloid leukemia [17]. *RASA3* hypomethylation is a frequent characteristic of hepatocellular carcinoma, and serves as a potential biomarker in early detection [18]. The CpG site cg13482620, which is located in *B3GNTL1*, was strongly associated with lung cancer in Norwegian women [19]. A *PIGB* mutation may lead to development and neurogenesis problems [20]. A *MAP3K5* mutation is related to malignant stages of prostate cancer [21]. A *LPIN1* mutation causes rhabdomyolysis [22]. A *MYBPC3* mutation was noted in patients with inherited hypertrophic cardiomyopathy [23]. Each gene plays a crucial role in different diseases. However, their function in CRC needs further research. On the basis of our selection, all of them are potential biomarkers.

Allele-specific DNA methylation of *SMAD3* is regulated based on genetic effects relevant to disease susceptibility. Single-nucleotide polymorphisms (SNPs), such as rs36221701, which are located upstream of *SMAD3*, are significantly related to gene expression in inflammatory bowel disease (IBD) [24], which may increase CRC risk by 3- to 5-fold. Crohn’s disease and ulcerative colitis are the principal types of IBD. SNP rs36221701 is considered a susceptibility locus in Crohn’s disease and is strongly associated with repeat operations [25]. Even among Western patients, *SMAD3* rs17293632 is a susceptibility locus in Crohn’s disease. Decreasing phosphorylation in *SMAD3* was observed in IBD, which may impair the immunosuppressive effect of TGF-β [26]. Therefore, we suggest that, in Western and Asian patients, *SMAD3* plays a vital role in the development of digestive disorders, which may increase the progression risk, leading to CRC.

DNA methylation of *SMAD3* was analyzed in 38 paired colorectal samples and 314 tumor sample datasets from TCGA. Results showed that significant hypomethylation occurs in tumor tissues on cg24032190 in *SMAD3*. Furthermore, we analyzed the DNA methylation level of *SMAD3* in 548 Taiwanese patients and nine polyp tubular adenomas. *SMAD3* hypomethylation was found in 91.4% (501/548) of Taiwanese CRC tissues, which is higher than the long interspersed nuclear element-1 (*LINE-1*) hypomethylation (66.2%) in CRC tissues and *SEPT9* hypermethylation (60.92%) in Taiwanese patients with CRC [27,28]. Furthermore, the Kaplan–Meier survival curves revealed nonsignificant differences between Taiwanese and TCGA datasets (Figure 8). In particular, Taiwanese male, elderly, and late-stage CRC patients without SMAD3 hypomethylation had a much better survival rate. To clarify the relevance of methylation level and survival rate in Western and Asian populations, increasing the sample size is necessary. Moreover, *SMAD3* hypomethylation was observed in 66.6% (6/9) of polyps, and the methylation level in polyps was between that of normal and tumor tissues (Figure 2B). The tendency of the methylation level can be determined based on the tissue condition. This may be used for confirming the precancerous condition before further investigation. On the basis of the outcome, *SMAD3* hypomethylation could be a superior early predictive biomarker in Asian patients with CRC.

We analyzed the mRNA expression of *SMAD3* in 119 Taiwanese patients and nine polyps through a reverse-transcription polymerase chain reaction (RT-PCR). Interestingly, *SMAD3* mRNA expression was low in 42.9% (51/119) of Taiwanese CRC tumor tissues and 33.3% of polyps (Figure 5B). This result might be caused by the hypermethylation of five CpG sites, which are located in promoter regions (Figure 2A). Consequently, our target CpG site cg24032190, which is located on the gene body, showed hypomethylation but low expression (Figure 6). We propose that promoter-site hypermethylation represses gene expression. Hypomethylation of the gene body does not have a major effect on gene expression [29]. Moreover, *SMAD3* deficiency promotes tumorigenesis in the distal colon of carrying an inactivated allele of the adenomatous polyposis coli gene (*Apc^Min/+^*) mice [30]. Deficient *SMAD3* expression is related to human gastric cancer [31]. *SMAD3* expression may play a crucial role in the carcinogenesis of the digestive system.

The circulating methylated level of *SMAD3* was examined through QMSP in 200 μL of plasma in 15 healthy participants and 15 patients with CRC. The outcome revealed decreased methylation in 86.6% of patients with CRC. However, 60% of healthy participants had false-positive results. Thus, circulating methylated *SMAD3* cannot distinguish patients from healthy participants in a plasma volume of 200 μL. As evidenced by previous studies, the column-based method may lead to cross-contamination [32,33]. To improve sensitivity (86.6%, 13/15), PPV (59%, 13/22), and experimental stability, we increased the plasma volume to 1 mL. Decreased methylation was detected in 70% (7/10) of patients with cancer but in 20% (2/10) of healthy individuals. The sensitivity is 78.5% (11/14) and PPV increased to 64.7% (11/17). Because of the limited number of samples available, the total plasma sample size was 58 individual specimens (30 participants for 200-μL plasma analysis and 28 for the 1-mL plasma analysis). Based on our results, we infer that *SMAD3* can be a noninvasive biomarker for detecting CRC. In addition, combining *SMAD3* with other hypomethylated genes, such as *ACOT7*, *RASA3*, *B3GNTL1*, *PIGB*, *MAP3K5*, *LPIN1*, and *MYBPC3*, which are potential biomarkers of CRC in the analytical model, may improve accuracy, increase sensitivity, and reduce false-positive results when predicting CRC in circulating methylated *SMAD3*. To confirm this, recruiting more patients with CRC and healthy participants is necessary.

We analyzed the methylation level of cg24032190 in *SMAD3* in different cancer types. However, no significant difference was observed in lung and breast cancer. Esophageal cancer is the only cancer that showed hypomethylation in Taiwanese patients (Table 2). By contrast, based on the TCGA data, cg24032190 *SMAD3* caused no significant difference in the methylation pattern in esophageal cancer (Figure 7 and Table 2). This indicates that *SMAD3* is a specific biomarker in Taiwanese esophageal cancer. In esophageal adenocarcinoma, cfDNA *LINE-1* hypomethylation is considered a possible molecular assay [34]. Therefore, further investigation to confirm the hypomethylation level of ccfDNA *SMAD3* in esophageal cancer is encouraged. After analyzing TCGA datasets, we found that *SMAD3* hypomethylation mainly occurs in the gastrointestinal tract. Whether aberrant *SMAD3* hypomethylation is associated with eating habits is also worthy of further study. Furthermore, we observed that cg24032190 of *SMAD3* is 60% hypermethylated in endometrial cancer in both Taiwanese patients and TCGA datasets (Table 2). Based on the previous paper, *SMAD3* can influence endometrial dysregulation and hormone-dependent uterine tumors [35].

In conclusion, *SMAD3* methylation levels vary in colorectal tissue based on the type of cancer. The methylation level in tissues indicates that *SMAD3* can be a potential biomarker for early prediction of CRC. However, the number of available plasma samples was limited. Further research will aim to increase the sample size to identify whether unmethylated *SMAD3* ccfDNA in plasma can predict a CRC precancerous condition before colonoscopy and biopsy. The trend in methylation level can be inferred from our results. Future studies should investigate whether measuring *SMAD3* from ccfDNA is useful in the case of early colonic disease.

## 4. Materials and Methods

### 4.1. Patients and Tissue and Plasma Collection

We obtained 548 paired adjacent normal colorectal tissues and CRC tissues, 16 paired esophageal cancer tissues, 33 paired lung cancer tissues, 15 paired endometrial cancer tissues, 23 paired breast cancer tissues, 29 healthy and 29 CRC plasma samples, and nine polyp tissues from Taipei Veterans General Hospital Biobank and Taipei Medical University Joint Biobank (Figure 9). Informed consent forms were signed by patients before specimen and clinical data collection. Patients undergoing preoperative chemoradiotherapy or an emergent operative procedure, who died within 30 postoperative days, or with evidence of familial adenomatous polyposis or Lynch syndrome, were excluded from this study.

The task of determining cancerous tumors and normal tissue was assigned to professional gastrointestinal pathologists. Personal clinical data and tumor conditions were supplied from the aforementioned two hospitals. After surgery, follow-ups are scheduled every three months for two years and semiannually thereafter. The follow-up protocol included physical examination, digital rectal examination, carcinoembryonic antigen analysis, chest radiography, abdominal sonogram, and computerized tomography, if required. Proton emission tomography or magnetic resonance imaging was arranged for patients with an elevated carcinoembryonic antigen level but tumor recurrence at an uncertain site.

### 4.2. Genomic DNA, ccfDNA, and RNA Extraction

All specimens were stored at −80 °C immediately after surgery. DNA from the paired tissue (cancerous and noncancerous tumor) from the same patient was extracted using the QIAamp DNA Mini Kit (Qiagen, Bonn, Germany; Cat. no. 51306). mRNA was extracted using the RNeasy Plus Mini Kit (Qiagen, Hilden, Germany; Cat. no. 74134). In 200 μL of plasma, ccfDNA was extracted using a MagMAX Cell-Free DNA Isolation Kit (Thermo Scientific, Austin, TX, USA; Cat. no. A29319). A total of 1 mL of plasma was extracted through automatic beads (Thermo Scientific, Waltham, MA, USA; Cat. no.100033590) by using a 24-well plate in a KingFisher^TM^ Duo Prime machine (Thermo Fisher Scientific, Inc., Woodlands, Singapore; Cat. no. 5400110). All the aforementioned processes were based on the manufacturer’s instructions and the recommended protocol.

### 4.3. Ethical Approval and Consent to Participate

The study was approved by the Taipei Medical University Joint Institutional Review Board and the Institutional Review Board, Taipei Veterans General Hospital. Written informed consent was obtained from all patients. The project identification codes are 201305002 and 2017-12-011CC, respectively.

### 4.4. Quantitative Reverse-Transcription Polymerase Chain Reaction (qRT-PCR)

LightCycler 480 (Roche Applied Science, Mannheim, Germany) was the main machine used to gauge mRNA expression and real-time RT-PCR of *SMAD3*. According to the manufacturer’s guideline, the LightCycler 480 Probe Master Kit (Roche Applied Science, Indianapolis, IN, USA; Cat. no. 04707494001) with specific primers and probe were used to perform real-time PCR. The glyceraldehyde 3-phosphate dehydrogenase (*GAPDH*) gene was the standard for comparison. The PCR conditions were as follows: 95 °C for 10 min and annealing temperature 60 °C for 10 s for a total of 45 cycles. In accordance with the instructions from the manufacturers, *GAPDH* was used as a reference gene. The normalized gene expression values obtained using LightCycler Relative Quantification software (version 1.5, Roche Applied Science) were then compared with those of the control group. The *SMAD3* mRNA expression level was considered high if the mRNA expression level of *SMAD3* was twice that of *GAPDH* in colorectal tumor tissue compared to normal colorectal tissue. Table 3 lists the primers.

### 4.5. TaqMan Quantitative Methylation-Specific PCR (QMSP)

The DNA methylation level of *SMAD3* was gauged using TaqMan QMSP with light cycler 480 (Roche Applied Science) after the bisulfite conversion with the EpiTect Fast DNA Bisulfite Kit (Qiagen, Bonn, Germany, cat. no. 59826), which was suggested by the manufacturer. To perform QMSP, a SensiFAST™ Probe No-ROX Kit (Bioline, London, UK; Cat. no. BIO-86020) with specific primers and probe was used for *SMAD3*. Normalized DNA methylation values from LightCycler Relative Quantification software (version 1.5, Roche Applied Science) were compared with the control group. The relative *SMAD3* DNA methylation level was normalized to Beta-actin (*ACTB*). *ACTB* can work as total genomic DNA or ccfDNA content control. *SMAD3* was considered hypomethylated when the methylation level of *SMAD3* relative to that of the *ACTB* gene was half that in CRC tissue compared with the paired noncancerous colorectal tissue sample. In circulating methylation, the average value of *SMAD3* relative to that of the *ACTB* gene in healthy tissue was 25.37. In CRC, a value of less than 0.5 (50-fold lower in CRC than healthy tissue) was regarded as hypomethylation. Table 3 presents the primers. The specificity of *SMAD3* methylation end products was confirmed by bisulfite sequencing (Appendix A).

### 4.6. Genome-Wide Methylation Analysis

The Illumina Infinium HumanMethylation450 BeadChip array (Illumina, San Diego, CA, USA) is widely used in various fields to measure DNA methylation. The array assessed more than 450,000 CpGs, and the coverage ratio of RefSeq genes is 99%. Thus, we used the Illumina Infinium HumanMethylation450 BeadChip array to perform genome-wide methylation analysis in 26 paired CRC tissues and adjacent noncancerous tissues. Based on the manufacturer’s instructions, the EpiTect Fast DNA Bisulfite Kit (Qiagen, Cat. no. 59826) was used to perform bisulfite conversion for 500 ng of genomic DNA per time. After calculating the sum of the methylated ratio, the methylation level of each CpG site was marked as “beta” for values ranging from 0 (unmethylated) to 1 (fully methylated).

### 4.7. Statistical Analysis

Pearson’s *X^2^* test was used to analyze *SMAD3* hypomethylation and mRNA expression in patients with CRC, and correlations with various clinical parameters, including age, sex, cancer type, stage, degree of differentiation, location, and microsatellite instability status were assessed. Nonparametric statistical tests were performed to compare DNA methylation and mRNA expression between polyps in normal and tumor tissue. Overall survival and cancer-specific survival were calculated and analyzed using the Kaplan‒Meier method. All statistical analyses were performed using SPSS software (Chicago, IL, USA). Pearson correlation and Spearman correlation were used to analyze the correlation between the DNA methylation and mRNA expression of *SMAD3*.

## Figures and Tables

**Figure 1 ijms-21-07395-f001:**
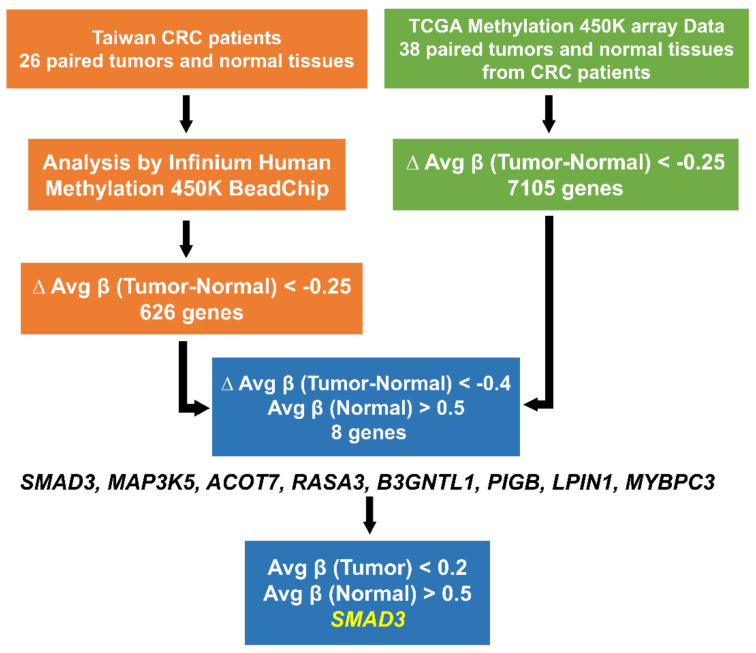
Criteria and step-by-step flowchart of gene selection.

**Figure 2 ijms-21-07395-f002:**
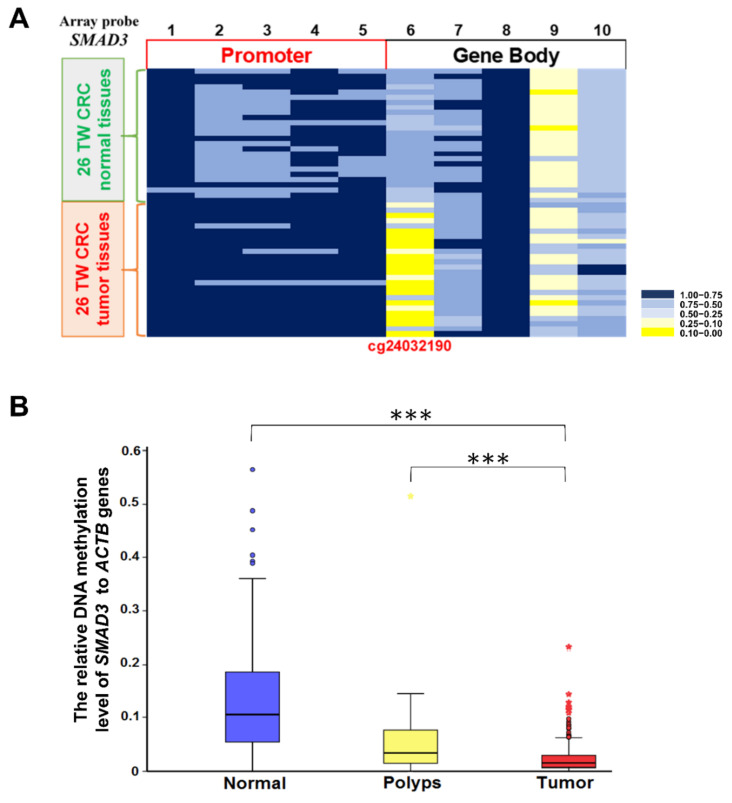
Methylation levels in Taiwanese patients with colorectal carcinoma (CRC). (**A**) Differentially methylated CpG heatmap of mothers against decapentaplegic homolog 3 (*SMAD3*) in 26 paired CRC patients. Methylation levels (average β values) at differentially methylated loci were identified using an Illumina Human Methylation 450K array-based assay. The five CpG sites in promoter regions −1133, −525, −413, −122, and −94 are designated 1, 2, 3, 4, and 5, respectively. The CpG sites in gene body regions +12535, +927, +11421, +1331, and +2511 are designated 6, 7, 8, 9, and 10, respectively. (**B**) Figures of the methylated *SMAD3* levels determined by quantitative methylation-specific polymerase chain reaction (QMSP) in 548 adjacent normal colon tissues, nine polyps of tubular adenoma, and 548 CRC tumors. Experiments were performed with three technical replicates. Results are shown in mean ± standard deviation. *** *p* ≤ 0.001. A *t*-test and nonparametric analysis was used to calculate group differences in all experiments.

**Figure 3 ijms-21-07395-f003:**
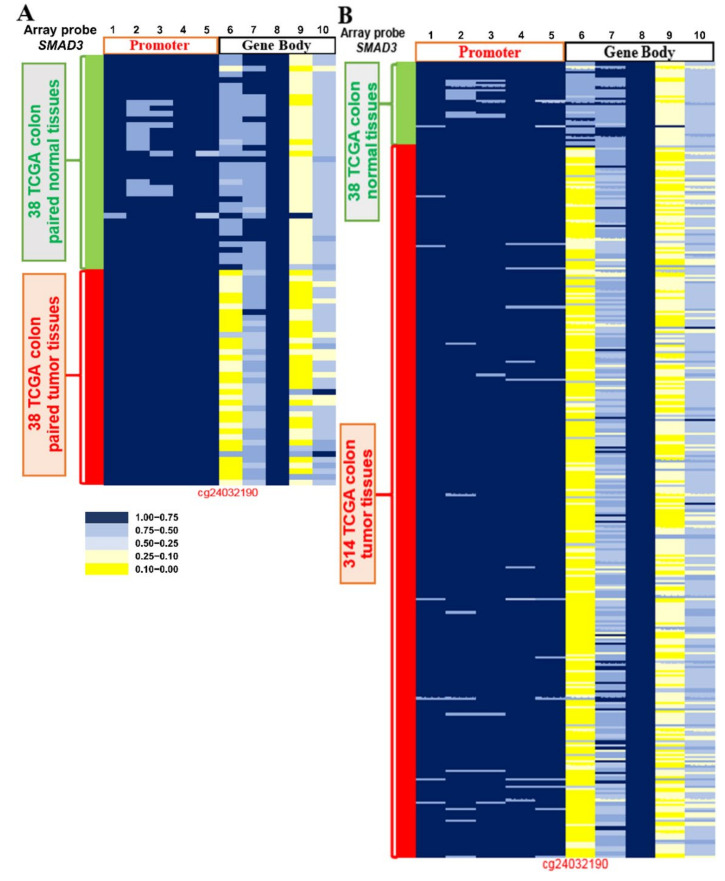
*SMAD3* DNA methylation analysis from The Cancer Genome Atlas dataset. Differentially methylated CpG sites in *SMAD3* were identified in (**A**) 38 adjacent normal colorectal tissues, 38 matched colorectal carcinoma (CRC) tumors, and (**B**) 314 CRC tumors by using an Illumina Human Methylation 450K array-based assay. The five CpG sites in promoter regions −1133, −525, −413, −122, and −94 are designated 1, 2, 3, 4, and 5, respectively. The CpG sites in gene body regions +12535, +927, +11421, +1331, and +2511 are designated 6, 7, 8, 9, and 10, respectively.

**Figure 4 ijms-21-07395-f004:**
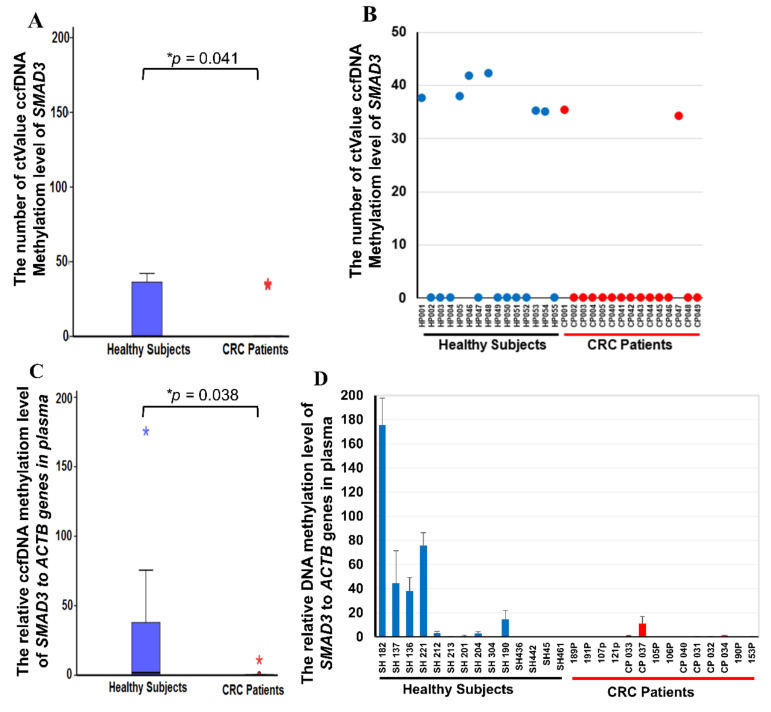
Circulating cell-free DNA methylation levels in Taiwanese patients with colorectal cancer (CRC). (**A**) The box plot of *SMAD3* methylation levels in 200 μL plasma. (**B**) Circulating methylated *SMAD3* levels determined by using quantitative methylation-specific polymerase chain reaction in 15 healthy subjects and 15 patients with CRC in 200 μL plasma extracted through a manual process. (**C**) The box plot of *SMAD3* methylation levels in 1 mL plasma. (**D**) Circulating methylated *SMAD3* levels in 14 healthy subjects and 14 CRC patients in 1 mL plasma. Experiments were performed with three technical replicates. * *p* ≤ 0.05. A *t* test was used to calculate group differences.

**Figure 5 ijms-21-07395-f005:**
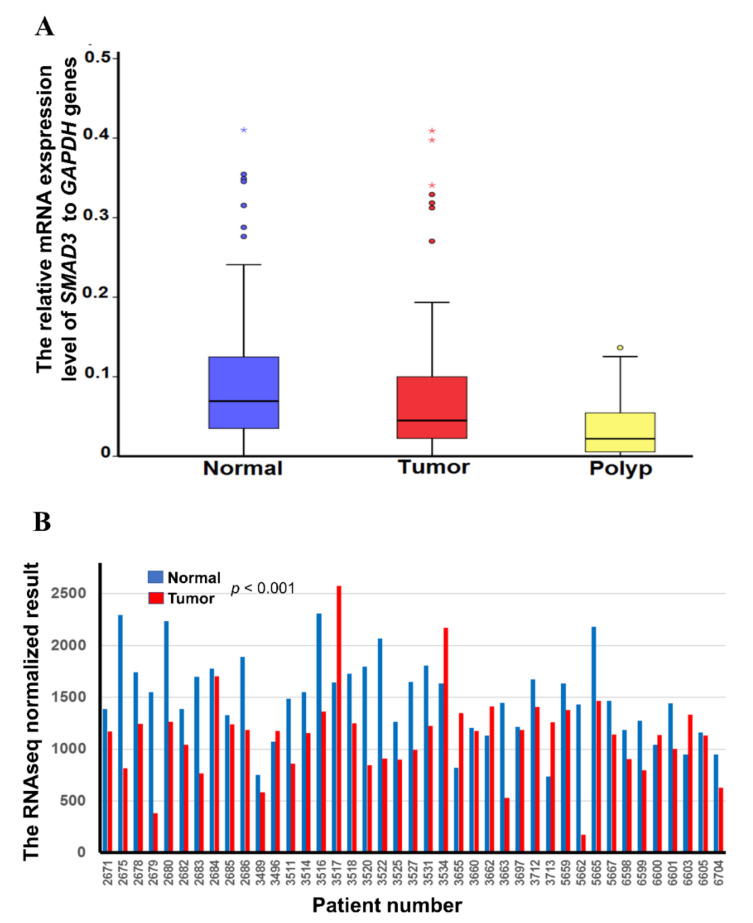
The mRNA expression in Taiwanese colorectal cancer (CRC) paired tissues, polyps, and The Cancer Genome Atlas (TCGA) dataset. (**A**) Boxplot of the *SMAD3* mRNA expression level determined by quantitative reverse transcription–polymerase chain reaction in 119 paired colon tissues and nine polyps from the Taiwanese population. (**B**) RNA sequencing data of *SMAD3* in 41 matched CRC tumors from the TCGA dataset. Results are shown as mean ± standard deviation.

**Figure 6 ijms-21-07395-f006:**
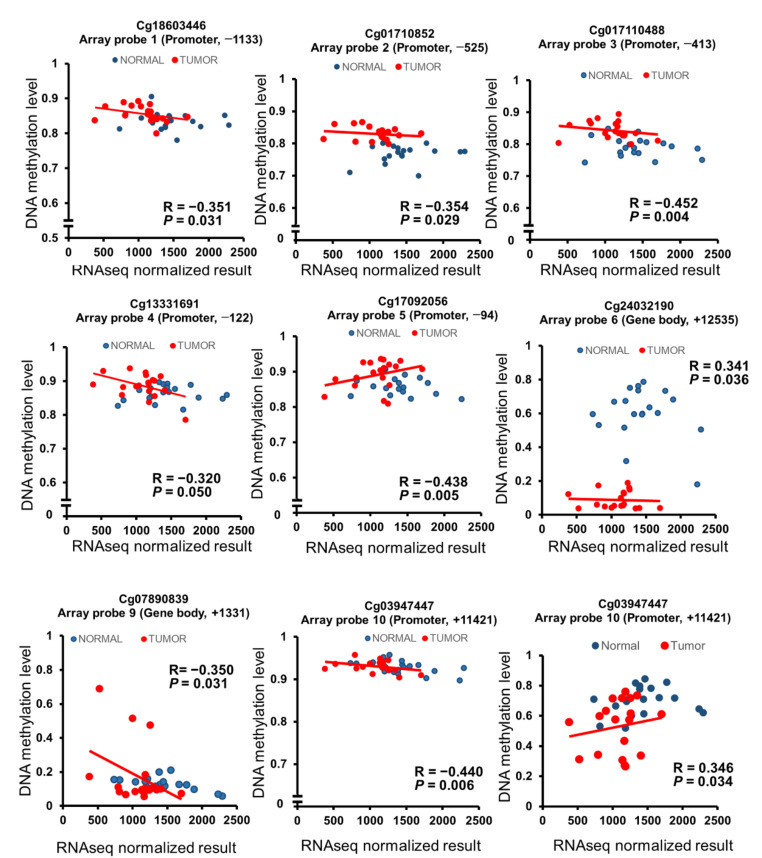
Pearson correlation analysis of tissues between DNA methylation and RNA sequencing in 19 paired adjacent normal tissue samples and 19 paired CRC tissue samples from patients.

**Figure 7 ijms-21-07395-f007:**
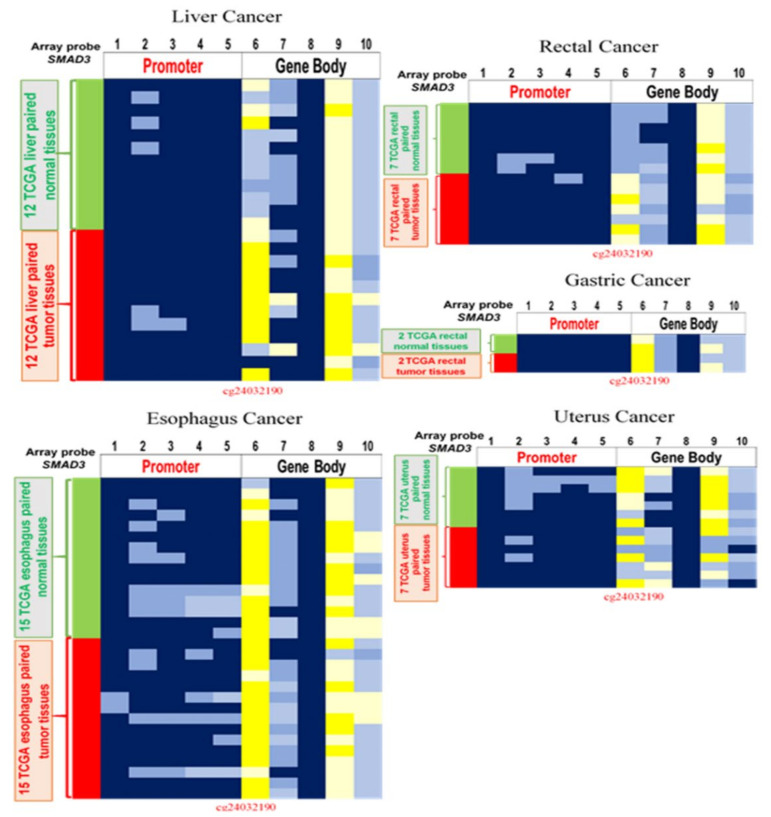
*SMAD3* DNA methylation in different cancers. Differentially methylated CpG heatmap of *SMAD3* in paired liver cancer, rectal cancer, gastric cancer, esophageal cancer, and uterine cancer. Methylation levels (average β values) at differentially methylated loci were identified by using an Illumina Human Methylation 450K array-based assay.

**Figure 8 ijms-21-07395-f008:**
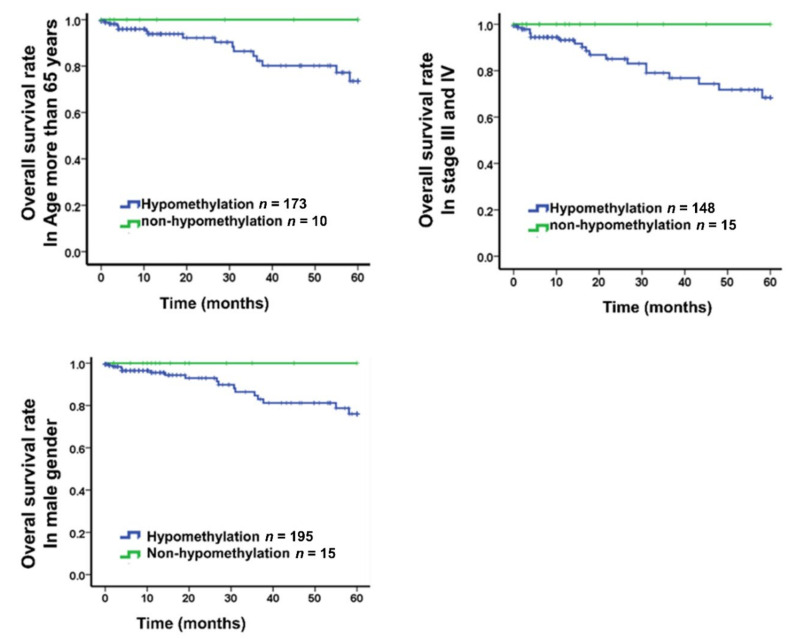
Kaplan–Meier survival curves were constructed to compare the overall survival between CRC patients with hypomethylation and nonhypomethylation of *SMAD3* in patients with age >65, stage III‒IV, and male gender. *SMAD3* was defined as hypomethylation when the methylation level in CRC tumors was half that in adjacent normal colorectal tissue.

**Figure 9 ijms-21-07395-f009:**
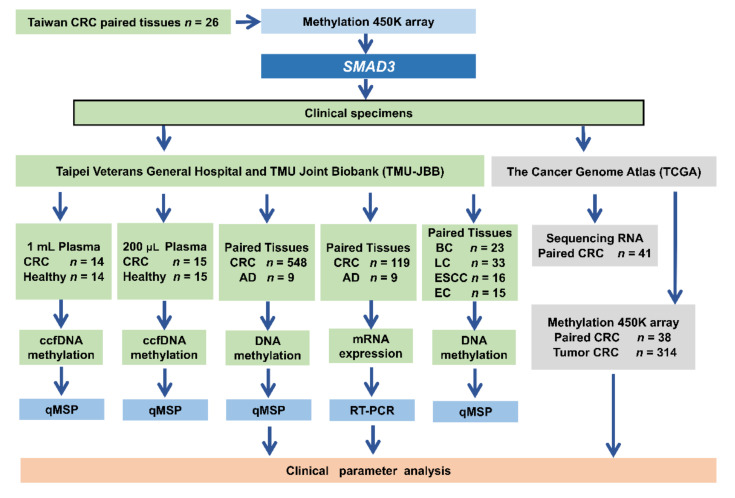
Flowchart of the study design, datasets and specimens used. For each step, the sample types and number of samples used for the analyses are indicated. CRC, colorectal cancer; AD, benign adenoma; BC, breast cancer; LC, lung cancer; ESCC, esophageal cancer; ES, endometrial cancer; ccfDNA, circulating cell-free DNA; QMSP, quantitative methylation-specific PCR; qRT-PCR, quantitative reverse-transcription PCR; methylation 450K array, Illumina Infinium HumanMethylation450 BeadChip array.

**Table 1 ijms-21-07395-t001:** Alterations of *SMAD3* in relation to the clinical parameters of colorectal cancer (CRC) in Taiwan.

Characteristics	Total *n*	*SMAD3* Methylation				Total *n*	*SMAD3* mRNA					
		Low *n* (%) ^a^		Normal *n* (%)			Low *n* (%) ^b^		Moderate *n* (%)		High *n* (%)	
**Overall**	548	501	(91.4)	47	(8.6)	119	51	(42.9)	33	(27.7)	35	(29.4)
**Age**												
<65	240	213	(88.8)	27	(11.2)	46	23	(50.0)	11	(23.9)	12	(26.1)
65	287	268	(93.4)	19	(6.6)	60	24	(40.0)	19	(31.7)	17	(28.3)
Sex												
Male	314	289	(92.0)	25	(8.0)	59	24	(40.7)	19	(32.2)	16	(27.1)
Female	214	193	(90.2)	21	(9.8)	54	26	(48.1)	13	(24.1)	15	(27.8)
**Tumor Type**												
Adeno	492	451	(91.7)	41	(8.3)	104	45	(41.7)	32	(29.6)	27	(25.0)
Mucinous	56	50	(89.3)	6	(10.7)	4	2	(50.0)	0	(0)	2	(50.0)
**Tumor Stage**												
0 and I	53	47	(88.7)	6	(11.3)	10	4	(40.0)	3	(30.0)	3	(30.0)
II, III, and IV	451	411	(91.1)	40	(8.9)	98	44	(44.9)	28	(28.6)	26	(26.5)
**Tumor Size**												
T0–T1	34	31	(91.2)	3	(8.8)	6	2	(33.3)	1	(16.7)	3	(50.0)
T2–T4	485	442	(91.1)	43	(8.9)	100	45	(45.0)	29	(29.0)	26	(26.0)
**Regional lymph nodes metastasis**												
N = 0	270	250	(92.6)	20	(7.4)	57	25	(43.9)	16	(28.1)	16	(28.1)
N ≥ 1	249	223	(89.6)	26	(10.4)	51	23	(45.1)	15	(29.4)	13	(25.5)
**Distant metastasis**												
M = 0	398	361	(90.7)	37	(9.3)	76	36	(47.4)	21	(27.6)	19	(25.0)
M ≥ 1	102	93	(91.2)	9	(8.8)	28	11	(39.3)	8	(28.6)	9	(32.1)
**Differentiation grade**												
Well/Moderate	481	439	(91.3)	42	(8.7)	101	45	(44.6)	30	(29.7)	26	(25.7)
Poor/undifferentiation	33	29	(87.9)	4	(12.1)	6	2	(33.3)	1	(16.7)	3	(50.0)
**Vascular invasion**												
No invasion	408	373	(91.4)	35	(8.6)	60	24	(40.0)	16	(26.7)	20	(33.3)
Invasion	77	70	(90.9)	7	(9.1)	18	8	(44.4)	3	(16.7)	7	(38.9)
**Location**												
Cecum, appendix	46	43	(93.5)	3	(6.5)	8	2	(25.0)	1	(12.5)	5	(62.5)
Ascending colon	88	83	(94.3)	5	(5.7)	13	4	(30.8)	4	(30.8)	5	(38.5)
Transverse colon	22	20	(90.9)	2	(9.1)	5	1	(20.0)	0	(0)	4	(80.0)
Descending colon	49	45	(91.8)	4	(8.2)	12	6	(50.0)	1	(8.3)	5	(41.7)
Sigmoid colon	160	142	(88.8)	18	(11.2)	23	12	(52.2)	6	(26.1)	5	(21.7)
Rectum	129	119	(82.2)	10	(7.8)	18	7	(38.9)	8	(44.4)	3	(16.7)
**MSI ^c^**												
MSS	50	47	(94.0)	3	(6.0)	38	12	(31.6)	14	(36.8)	12	(31.6)
MSI-L	6	6	(100.0)	0	(0.0)	5	2	(40.0)	2	(40.0)	1	(20.0)
MSI-H	9	9	(100.0)	0	(0.0)	7	4	(57.1)	0	(0)	3	(42.9)

These results were analyzed based on the Pearson *X*^2^ test. For some categories, the number of samples (*n*) was lower than the overall number analyzed because clinical data were unavailable for those samples. ^a^ The *SMAD3* methylation level in CRC tumors was half that in adjacent normal colorectal tissues—defined as hypomethylation. ^b^ The *SMAD3* expression level in CRC tumors was half that in adjacent normal colorectal tissues—defined as low expression. ^c^ MSI: microsatellite instability.

**Table 2 ijms-21-07395-t002:** Methylation level of *SMAD3* in different cancers.

	TCGA				TAIWAN						
	Total *n*	Avgβ (T)	Avgβ (N)	Avgβ (T − N)		Total *n*	Avg (T)	Avg (N)	Pair T/N ^a^ < 0.5	Pair T = N	Pair T/N > 2
**Colon**	38	0.11	0.63	−0.52	**Colon**	548	0.230	0.010	91.4%	3.6%	5.4%
**ESCC ^b^**	15	0.06	0.09	−0.03	**ESCC**	16	0.002	0.001	62.5%	25%	12.5%
**Breast**	87	0.13	0.11	0.02	**Breast**	23	0.020	0.010	30.4%	47.8%	21.7%
**Gastric**	2	0.06	0.13	−0.07	**Lung**	33	0.002	0.003	42.4%	33.3%	24.4%
**Liver**	12	0.13	0.28	−0.15	**Endometrial**	15	0.003	0.020	0%	40%	60%
**Lung AD ^c^**	29	0.17	0.19	−0.02							
**Lung SQ** **^d^**	40	0.10	0.09	0.01							
**Pancreatic**	10	0.07	0.08	−0.01							
**Uterine**	7	0.32	0.13	0.17							

^a^ The results of the Pair T/N (tumor tissue/normal tissue) ratio were calculated through quantitative methylation-specific polymerase chain reaction analysis in tumor compared with adjacent normal tissues of patients with cancer. Pair T/N ratio < 0.5 was defined as hypomethylation. Pair T/N ratio > 2 was defined as hypermethylation. ^b^ ESCC: Esophageal Squamous Cell Cancer. ^c^ Lung AD: Lung Adenocarcinoma. ^d^ Lung SQ: Lung Squamous Cell Carcinoma.

**Table 3 ijms-21-07395-t003:** List of primer sequences and their reaction conditions used in the present study.

		Sequence (5′ to 3′)	Probe	Tm (°C)	Size (bp)
Real-time RT-PCR^a^					
GAPDH	Forward	AGCCACATCGCTCAGACAC	#60	60	66
	Reverse	GCCCAATACGACCAAATCC			
SMAD3	Forward	GTCTGCAAGATCCCACCA	#79	59	88
	Reverse	AGCCCTGGTTGACCGACT			
**Quantitative Methylation-Specific PCR**					
ACTB	Forward	TGGTGATGGAGGAGGTTTAGTAAGT		60	132
	Reverse	AACCAATAAAACCTACTCCTCCCTTAA			
	TagMan probe	ACCACCACCCAACACACAATAACAAACACA			
SMAD3	Forward	GAATAAGGTCGTTAGTTATTATCGT		54.48	172
	Reverse	AATCAAATCTACCCGAATCGAA			
	TaqMan probe	GAAAGAAAGAAAGAAAGTAAATTTTATTTTTAAGCG			

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
