# Peer review of "SMAD3 Hypomethylation as a Biomarker for Early Prediction of Colorectal Cancer"

_ijms, 2020, doi:10.3390/ijms21197395_

Round 1

Reviewer 1 Report

I would like to thank the authors for addressing my initial comments. Following the revision to the article, some of my additional comments concern only clarity in the interpretation of results.

In the legend of Figure 3 the authors may want to introduce the location for the probes 1 to 10 (as they have done for Figure 2). Also, in Figure 3A please correct “Array probe SMAD3” (instead of Arraya probe SMAD3).

page 6 the phrase “A decrease in methylation was detected in 86.6% (13/15) of patients with CRC but in …” the word “but” should be replaced.

Please clarify: Why in these analyses the authors only used 15 subjects? In the Mat & Met section the authors present 25 healthy and 25 CRC plasma samples. Then in Figure 4 panel D the authors present 14 samples, are those different patients than the 15 from panel B?

Page 7- Figure 5A, the authors present in the text the p values for the analysis between normal tissues and tumor tissues and between tumor tissues and polyps? The text should be re-written for clarification.

Moreover, regarding the Figure 5B the authors say “Low expression (p≤0.001) was found in TCGA datasets” This phrase needs to be clarified for better understanding; Low expression in the tumor tissue versus the normal?

In Figure 6, please clarify the legend of the Figure. “…19 paired-tissues (tumor and normal adjacent mucosa)…” is this what the authors mean by paired adjacent normal and 19 paired patients with colorectal cancer?

Again, why in this analysis were used 19 patients? The authors should discussion the biological meaning of these correlations described for figure 6.

Author Response

Response to Reviewer 1 comments:

We would like to thank reviewer 1 for taking the time and effort necessary to review the manuscript. We sincerely appreciate all valuable comments and suggestions, which helped us to improve the quality of the manuscript.

Point 1: In the legend of Figure 3 the authors may want to introduce the location for the probes 1 to 10 (as they have done for Figure 2). Also, in Figure 3A please correct “Array probe SMAD3” (instead of Arraya probe SMAD3)

Response 1: We apologize for the error. The legend and figure have revised accordingly. (page 5, line 148-150)

Point 2: Page 6 the phrase “A decrease in methylation was detected in 86.6% (13/15) of patients with CRC but in …” the word “but” should be replaced.

Response 2: As suggested by the reviewer, the word “but” has been replaced with “and”. (Page 6, line 155)

Point 3: Why in these analyses the authors only used 15 subjects? In the Mat & Met section the authors present 25 healthy and 25 CRC plasma samples. Then in Figure 4 panel D the authors present 14 samples, are those different patients than the 15 from panel B?

Response 3: We apologize for the mistake and have corrected it. The samples from 14 healthy people and 14 CRC patients in 1 mL are different from the samples in 200 μL. In 200 μL plasma, we have 15 healthy and 15 CRC samples. Consequently, the total number of plasma samples are 29 healthy and 29 CRC samples. (Page 14, line 349)

Point 4: Page 7- Figure 5A, the authors present in the text the p values for the analysis between normal tissues and tumor tissues and between tumor tissues and polyps? The text should be re-written for clarification.

Response 4: We thank the reviewer for the suggestions and comments. The sentence has been rewritten to “No significant difference was found between normal tissues and polyps, or between tumor tissues and polyps, in terms of mRNA expression”. (Page 7, line 175-176)

Point 5: Moreover, regarding the Figure 5B the authors say “Low expression (p≤0.001) was found in TCGA datasets” This phrase needs to be clarified for better understanding; Low expression in the tumor tissue versus the normal?

Response 5: We agree with the reviewer’s comment. Accordingly, we have corrected the sentence to “Low expression in the tumor versus normal tissue was found in TCGA datasets (p ≤ 0.001)”. (Page 7, line 179-180)

Point 6: In Figure 6, please clarify the legend of the Figure. “…19 paired-tissues (tumor and normal adjacent mucosa)…” is this what the authors mean by paired adjacent normal and 19 paired patients with colorectal cancer?

Response 6: In this manuscript, the adjacent nontumor part was collected from the colonic mucosa 10 cm proximal from the main tumor. It may include mucosa, submucosa, muscularis propria (externa) and serosa (perimuscular tissue in rectum). In order to make the description clear, we have modified the sentence to “19 paired adjacent normal tissue samples and 19 paired CRC tissue samples from patients”. (Page 9, line 200)

Point 7: As mentioned above, why in this analysis were used 19 patients? The authors should discussion the biological meaning of these correlations described for figure 6.

Response 7: In order to keep the accuracy of study. We excluded the missing data from TCGA datasets. According to the data we have, only 19 patients have the completely data (RNA sequence normalized result and DNA methylation level).

Reviewer 2 Report

In this study, the authors by using a genome-wide methylation analysis identify a hypomethylation site in an intron of the SMAD3 gene. By quantitative methylation-specific polymerase chain reaction (QMSP) they demonstrated that about 91% of Taiwanese patients with CRC showed SMAD3 hypomethylation. This finding was in agreement with SMAD3 hypomethylation observed in Western patients with CRC from The Cancer Genome Atlas dataset. In addition, Hypomethylated SMAD3 was found in some cancers of the digestive system such as liver cancer, gastric cancer. The authors examined by QMSP also the circulating methylation level of SMAD3 in 200 μL (n= 15) and in 1ml (n= 14) of plasma extracted manually from patients with CRC and healthy participants. They detected decrease in circulating cell free methylation of SMAD3 in 86% of CRC patients and in 60% of healthy individuals, starting from 200 μL of plasma, and a decrease in 70% of CRC patients and in 20% of healthy individuals, starting from 1ml of plasma.

In my opinion the topic of the paper is of scientific and clinical relevance for scientific community and clinical practices.  However, in my opinion some revisions are needed before the paper could be published.

  • The authors report having analyzed TCGA Illumina Infinium HumanMethylation450 BeadChip array data from 38 pairs of Western cancerous and noncancerous tissues (lines 87-88) but in figure 1 are reported 37 paired tumors and normal.
  • They report that q-RT-PCR conditions were 95 °C for 10 min and 60 °C for 10 min for a total of 45 cycles (line 377). 10 minutes of annealing/extension is vastly longer than that required to amplify a stretch of DNA smaller than 100 bp and, in such conditions, there would be the risk of amplifying non-specific products. Furthermore, a protocol that includes 10 minutes of denaturation at each cycle could result in the inactivation of the taq DNA polymerase after a few amplification cycles. They must clarify this point.
  • It is not clear the method that the authors used to quantify SMAD3 mRNA expression level (lines 380-381). I suggest that the authors explain it in detail.

  • In Table 3 they reported TagMan instead of TaqMan

  • In the section 4.4 of material and methods -TaqMan Quantitative Methylation-Specific PCR (QMSP)- the authors report having performed a relative quantification of SMAD3 methylation level (lines 390-393). However, the graphs of figure 4 that illustrate circulating cell-free DNA methylation levels in Taiwanese patients with colorectal cancer, are not at all clear. The authors must explain it in detail, alternatively, as the data concerning the circulating methylation level of SMAD3 are not consistent, they could eliminate this part from the paper.

  • I recommend the Author to carefully read the paper to revise some language misspellings.

Author Response

Response to Reviewer 2 comments:

We would like to thank reviewer 2 for taking the time and effort necessary to review the manuscript. We sincerely appreciate all valuable comments and suggestions, which helped us to improve the quality of the manuscript.

Point 1: The authors report having analyzed TCGA Illumina Infinium HumanMethylation450 BeadChip array data from 38 pairs of Western cancerous and noncancerous tissues (lines 87-88) but in figure 1 are reported 37 paired tumors and normal.

Response 1: We apologize for the error. The figure 1 has been modified to “38 paired tumors and normal tissue from CRC patients”. (Page 3, figure 1)

Point 2: They report that q-RT-PCR conditions were 95 °C for 10 min and 60 °C for 10 min for a total of 45 cycles (line 377). 10 minutes of annealing/extension is vastly longer than that required to amplify a stretch of DNA smaller than 100 bp and, in such conditions, there would be the risk of amplifying non-specific products. Furthermore, a protocol that includes 10 minutes of denaturation at each cycle could result in the inactivation of the taq DNA polymerase after a few amplification cycles. They must clarify this point.

Response 2: Thanks for your careful reading of our manuscript. We apologize for the typing error. The criteria of annealing/extension process should be 60 °C for 10 s. We have corrected the manuscript. (Page 15, line 386)

Point 3: It is not clear the method that the authors used to quantify SMAD3 mRNA expression level (lines 380-381). I suggest that the authors explain it in detail.

Response 3: We thankful the reviewer for the suggestions and comments. We have clarified the description to “In accordance with the instructions from the manufacturers, GAPDH was used as a reference gene. The normalized gene expression values obtained using LightCycler Relative Quantification software (version 1.5, Roche Applied Science) were then compared with those of the control group. The SMAD3 mRNA expression level was considered high if the mRNA expression level of SMAD3 was twice that of GAPDH in colorectal tumor tissue compared to normal colorectal tissue. Table 3 lists the primers”. (Page 16, line 390-395)

Point 4: In Table 3 they reported TagMan instead of TaqMan

Response 4: We apologize for the error and have corrected it to “TaqMan”. (Page 16, Table 3)

Point 5: In the section 4.4 of material and methods -TaqMan Quantitative Methylation-Specific PCR (QMSP)- the authors report having performed a relative quantification of SMAD3 methylation level (lines 390-393). However, the graphs of figure 4 that illustrate circulating cell-free DNA methylation levels in Taiwanese patients with colorectal cancer, are not at all clear. The authors must explain it in detail, alternatively, as the data concerning the circulating methylation level of SMAD3 are not consistent, they could eliminate this part from the paper.

Response 5: Thank you for your kind suggestions. We have re-written the sentence as “The relative SMAD3 DNA methylation level was normalized to ACTB. ACTB can work as total genomic DNA or ccfDNA content control. SMAD3 was considered hypomethylated when the methylation level of SMAD3 relative to that of the ACTB gene was half that in CRC tissue compared with the paired noncancerous colorectal tissue sample. In circulating methylation, the average value of SMAD3 relative to that of the ACTB gene in healthy tissue was 25.37. In CRC, a value less than 0.5 (50-fold lower in CRC than healthy tissue) was regarded as hypomethylation.”. (Page 16, line 407-412)

Point 6: Recommend the Author to carefully read the paper to revise some language misspellings.

Response 6: Thanks for your careful reading of our manuscript. We have proofread the manuscript by experienced scholarly writers who are native speakers from Wallace Academic Editing and MDPI English Editing, and the mistakes have been modified.

This manuscript is a resubmission of an earlier submission. The following is a list of the peer review reports and author responses from that submission.

Round 1

Reviewer 1 Report

Ansar et al report the the identification of hypomethylation of a CpG within SMAD3 gene in >90% of CRCs from TCGA and Taiwanese datasets and also in other cancers of the digestive tract.  SMAD3 hypomethylation was evaluated ccfDNA from CRC affected and healthy people and found to be decreased in 70% of CRC and 20% of healthy people.

The authors should discuss potential reasons why only 1 CpG in the SMAD3 gene region showed significant hypomethylation when often gene regions show concordant levels of methylation disruption across multiple adjacent CpGs? 

The concluding sentences of discussion should be reworded.  The sentence "SMAD3 as a potential noninvasive biomarker for early prediction of CRC" isn't supported by the results.  Although there was a significant difference in sMAD3 methylation between CRC-affected and healthy people in ccfDNA from plasma, more than half of the healthy people showed similar results to CRC-affected (Figure 4B) so would expect the positive predictive value to be low. therefore the concluding sentences are overstated.

It would have been useful to evaluate SEPT9 in parallel with SMAD3 in this study so that SMAD3 could be evaluated against a current biomarker, which would be achievable from the array data.

Measuring the SMAd3 methylation in ccfDNA from people with adenoma would have also added to determining the utility of this hypomethylation as a biomarker of early colonic disease.

Author Response

  1. The authors should discuss potential reasons why only 1 CpG in the SMAD3 gene region showed significant hypomethylation when often gene regions show concordant levels of methylation disruption across multiple adjacent CpGs?

Response 1:

The represented CpG site (region +12535, array probe 6) in the Illumina methylation array was commercially selected. Not all CpG sites can be detected with an Illumina methylation array because of its limitation to 450,000 CpG methylation sites. However, a specific methylated primer design and sequencing revealed that several CpG sites near region +12535 showed decreased methylation in CRC tumor tissues. The result has been added to the Results section (page 3, lines 106–110). The verified bisulfite sequencing is presented as follows (Supplementary Figure S3).

  1. The concluding sentences of discussion should be reworded. The sentence "SMAD3 as a potential noninvasive biomarker for early prediction of CRC" isn't supported by the results. Although there was a significant difference in SMAD3 methylation between CRC-affected and healthy people in ccfDNA from plasma, more than half of the healthy people showed similar results to CRC-affected (Figure 4B) so would expect the positive predictive value to be low. Therefore, the concluding sentences are overstated.

Response 2:

We thank the reviewer for the suggestions and comments. The concluding sentences in the Discussion section have been rewritten (page 14, lines 308–311). In Figure 4B, a significant difference (p = 0.041) was noted with 200 μL of plasma, but the positive predictive value (59%, 13/22) was low. Therefore, we increased the volume of plasma to 1 mL (Figure 4D), which resulted in a higher positive predictive value (64.7%, 11/17). These results imply that unmethylated ccfDNA in SMAD3 can predict early CRC from plasma. We will increase the sample size to prove our hypothesis in a future study (page 14, lines 306–309).

  1. It would have been useful to evaluate SEPT9 in parallel with SMAD3 in this study so that SMAD3 could be evaluated against a current biomarker, which would be achievable from the array data.

Response 3:

We performed SEPT9 methylation analysis on Taiwanese patients with CRC in a previous study by using a human methylation 450K array. The alteration rate of SEPT9 was found to be 60.92% in the Taiwanese population, whereas that of SMAD3 was much higher (91.4%). This comparison has been added to the Discussion section (page 13, lines 279–280).

  1. Measuring the SMAD3 methylation in ccfDNA from people with adenoma would have also added to determining the utility of this hypomethylation as a biomarker of early colonic disease.

Response 4:

In accordance with our result, the methylation level of polyps (Figure 2B) is between that of tumor and normal tissues. Consequently, predicting early colonic disease in ccfDNA from people with adenoma can be challenging. We have added the following text to the Discussion section (page 14, lines 329–334): “The methylation level in tissues indicates that SMAD3 can be a potential biomarker for early prediction of CRC. However, the number of available plasma samples was limited. Further research will aim to increase the sample size to identify whether unmethylated SMAD3 ccfDNA in plasma can predict a CRC precancerous condition before colonoscopy and biopsy. The trend in methylation level can be inferred from our results. Future studies should investigate whether measuring SMAD3 from ccfDNA is useful in the case of early colonic disease”.

Reviewer 2 Report

The authors analysed the SMAD3 hypomethylation and mRNA expression, in colorectal cancer specimens respect to normal mucosa, demonstrating and overexpression of SMAD3 in tumors and a hypomethylation.

The study is interesting, but the case series is very limited, and some points should be improved:

- In the abstract the authors should rewrite the last sentence when they say that SMAD3 hypomethylation is commonly observed in Western and Asian populations.

- Regarding RAS mutation, it has not been specified if these mutations have been analysed in these patients. Moreover, NRAS gene should be analysed together with KRAS.

- It is no clear why it was chosen 15 samples of the healthy subjects and if these samples were only used for ccfDNA extraction. It will be good if the authors elaborate a little why there was a need to obtain these samples from these healthy individuals.

- the authors considered the different tumor location in the analysis. It could be interesting to group tumors in right- versus left- sided tumors. What the authors mean by Others in the tumor location?

-  Discussion section should be improved

- The overall study is well balanced, but the investigators should make better use of appropriate statistical tools for the analysis. The sample size is small in some cases and in some can be considered not enough for this study. For instance, in Table 1 the authors say that “significant p values are indicated using superscripts” but this is not visible in the table, is there no significant p value? The authors could put in a column the p values.

The investigators use different number of samples for different analysis and it´s difficult to follow why exactly was the reason for this.

The authors in the mat & methods refer that “clinical data will be updated every 3 months for 2 years. Furthermore, the frequency will change to every 6 months in the next 3 years” what the authors mean by this?

 The authors have evaluated the prognosis and carried out the survival studies meaning the study spanned over several years and in such studies, it is important if not essential to mention the dates and overall time period of the study. Moreover, why these results are show in the discussion section instead of Results?

- pag 3 line 100 is missing a reference when the authors say that “SMAD3 promotes cancer progression through the TGF-β signalling pathway”.

-pag 3 line 107, when the authors say that “showed a significant hypomethylation difference” they should present that difference and how it was calculated.

- The authors should clarify the exact location of the CpG sites instead of only say “gene body”. It is helpful to identify the CpG sites in the figures instead of putting 1, 2, 3, 4, 5, 6, 7, 8, 9, 10.

- In Figure 2B only appear 19 patients, why? In the legend they say 20, this must be corrected. Again, in the text the authors only refer percentages, they should present statistical analysis.

Then in the QMSP the authors only used 10 samples, why?

- pag 5, line 133, the phrase should be re-phrased.

- pag 6 the text under the point 2.4 need to be better explained. For example, what the authors mean by the part extracted using automatic beads in 1 mL of plasma?

The text only refers to Figure 4 A and 4B, what about the results from figure 4C and 4D?

- There are some symbols used in the figure 4A and 4C that are not explained in the figure legend.

Why in these analyses the authors only used 15 subjects?

- Again, statistical analysis should be performed and presented, what the 20% in healthy means?

- the results in the text pag 7 lines 156 to 158 need to be modified. Is there a two-fold decrease or increase in SMAD3 expression in CRC tissues versus normal tissues?

When the authors say insignificant difference was observed, they should present the values and how they were obtained.

The authors when they refer to the results of mRNA expression of SMAD3 in the polyps samples they say 3/9 showed a low mRNA, is this which ones? because in the figure it seems that there´s low expression in polyps 2, 3, 4 and 5. Please explain. Also the statistical analysis is needed.

- The authors say “the trend of low expression…” did the authors done any statistical analysis to show if there is a difference between the expression levels in normal vs tumoral tissues using the TCGA data? If so the authors should mention it.

- In the figure 5, the authors should think about a way to present the data using all the patients and not just show part. Maybe if they calculate the difference between the expression for each paired sample and then use this to statistically analysed in all the samples if there is a difference in fold change.

- When the authors say: “ we found that hypermethylation of five CpG sites that are located at promoter regions cause low mRNA expression” the authors should also mention the two CpG sites in the gene body region that also give a correlation, again the location and identification of each probe would be positive for understanding better the results.

- In the figure 3 the different colors should be identified in the legend.

- In the figure 16: “19 normal controls and 19 paired patients with colorectal cancer” please explain better, which ones are the 19 normal controls?? Are they the paired of the 19 CRC samples or are those other samples?

- pag 9 line 187, please rephrase: “SMAD3 can be detected not only in the late stage but also in the early stage” Please explain referring to the data obtained.

- pag 9 line 188, please rephrase “SMAD 3 widely exists”

- pag 9, line 189: “SMAD3 is a potential biomarker for early prediction of CRC” must be better explained, how the authors reach this hypothesis?

- In the Table 1 (and table S1) only some characteristics from the tables are described in the text, while others are not such as: vascular invasion, location, MSI…

- The authors have checked for methylation levels of SMAD3 in other cancers, from where did these samples came? For some of these the numbers are too low to speculate about the methylation levels, for instance they only have 2 samples from gastric cancer, 7 from rectal cancer and 12 from liver cancer.

- page 12, line 258, what the authors mean by invasive investigation?

- page 13, line 270, what the authors mean by “this phenomenon confirms that hypermethylation in promoter sites repress gene expression”.

- The authors mention that using 200uL of plasma the healthy participants showed false-positive results, it was not clear if this problem is no longer a problem when they used 1mL of plasma.

- Table 2 is presented in the discussion section, what is the reason for this? Some tumors have really low number, did the authors done statistical analysis using these data?

- line 353: “mRNA expression level of SMAD3 relative to GAPDH was two-fold higher” please explain this sentence better.

There are various places which require grammatical correction. I have highlighted a few of these.

  1. a) 306 invasive check, replace this term
  2. b) the authors use the term “control group”, “patients”, “cases”, “cohort”, “specimens” these should be uniformized.

Author Response

  1. In the abstract the authors should rewrite the last sentence when they say that SMAD3 hypomethylation is commonly observed in Western and Asian populations.

Response:

We thankful the reviewer for the suggestions and comments. The last sentence in the abstract has been rewritten as follows: “In conclusion, SMAD3 hypomethylation is a potential diagnostic marker for CRC in Western and Asian populations.” On the basis of the results of tissue methylation levels, the incidence rate of hypomethylated SMAD3 in TCGA datasets and Taiwanese patients with CRC is 94.7% and 91.4%, respectively (page 1, lines 39–40).

  1. Regarding RAS mutation, it has not been specified if these mutations have been analysed in these patients. Moreover, NRAS gene should be analyzed together with KRAS

Response:

Because of limitations of the available datasets, we could not obtain the NRAS mutation data from Western and Taiwanese CRC patients.

  1. It is no clear why it was chosen 15 samples of the healthy subjects and if these samples were only used for ccfDNA extraction. It will be good if the authors elaborate a little why there was a need to obtain these samples from these healthy individuals.

Response:

The limited number of available plasma samples is a limitation of this study. This has been added as a study limitation in the Discussion section (page 14, lines 329–332): “The methylation level in tissues indicates that SMAD3 can be a potential biomarker for early prediction of CRC. However, the number of available plasma samples was limited. Further research will aim to increase the sample size to identify whether unmethylated SMAD3 ccfDNA in plasma can predict a CRC precancerous condition before colonoscopy and biopsy.”

  1. The authors considered the different tumor location in the analysis. It could be interesting to group tumors in right- versus left- sided tumors. What the authors mean by Others in the tumor location?

Response:

We also analyzed the correlations between SMAD3 alterations and various tumor locations: the cecum, appendix, ascending colon, transverse colon, descending colon, sigmoid colon and rectum. SMAD3 alterations were observed to be high in all locations. These data have been added to Table 1 in the Results section (page 10).

Characteristics

Total n

SMAD3 Methylation

  Low n(%) Normal n(%)

Total

n

  SMAD3 mRNA

Low n(%) Moderate n(%) High n(%)

Location

494

452

(91.5)

42

(8.5)

79

32

(40.5)

20

(25.3)

27

(34.2)

Cecum, appendix

46

43

(93.5)

3

(6.5)

8

2

(25.0)

1

(12.5)

5

(62.5)

Ascending colon

88

83

(94.3)

5

(5.7)

13

4

(30.8)

4

(30.8)

5

(38.5)

Transverse colon

22

20

(90.9)

2

(9.1)

5

1

(20.0)

0

(0)

4

(80.0)

Descending colon

49

45

(91.8)

4

(8.2)

12

6

(50.0)

1

(8.3)

5

(41.7)

Sigmoid colon

160

142

(88.8)

18

(11.2)

23

12

(52.2)

6

(26.1)

5

(21.7)

Rectum

129

119

(92.2)

10

(7.8)

18

7

(38.9)

8

(44.4)

3

(16.7)

  1. In Table 1, the authors say that “significant p values are indicated using superscripts” but this is not visible in the table, is there no significant p value? The authors could put in a column the p values.

Response:

We apologize for the incorrect description in the table footnotes. Comparison of SMAD3 hypomethylation levels with clinical parameters in Taiwanese patients revealed nonsignificant p values in clinical characteristics (Table 1) due to the extremely high frequency of hypomethylated SMAD3 in patients with CRC in Taiwanese and TCGA datasets. Only in Western TCGA datasets did CRC with MSI indicate significant differences in high mRNA expression (p = 0.02) (Table S1). Corrections have been made accordingly to Tables 1 (page 10) and S1.

  1. The investigators use different number of samples for different analysis and it´s difficult to follow why exactly was the reason for this.

Response:

Because of the limited number of available plasma samples, we tried to analyze all available samples. The exact number of specimens used in each experiment is indicated in the flowchart of the study design (page 15, Figure 9).

  1. The authors in the mat & methods refer that “clinical data will be updated every 3 months for 2 years. Furthermore, the frequency will change to every 6 months in the next 3 years” what the authors mean by this?

Response:

According to clinical guidelines in Taiwan, follow-ups are scheduled every 3 months for 2 years and semiannually thereafter. The description has been modified accordingly (page 15, lines 354–355).

  1. The authors have evaluated the prognosis and carried out the survival studies meaning the study spanned over several years and in such studies, it is important if not essential to mention the dates and overall time period of the study. Moreover, why these results are show in the discussion section instead of Results?

Response:

Indeed, a prognostic study is critical. However, the correlation between SMAD3 hypomethylation and survival time is unclear due to the small non-hypomethylated SMAD3 group. A description has been added to the Results section (page 12, point 2.8), and this point has been added as a study limitation in the Discussion section (page 13, lines 280–282).

  1. pag 3 line 99 is missing a reference when the authors say that “SMAD3 promotes cancer progression through the TGF-β signaling pathway”.

Response:

We apologize for the error and have added a reference to the sentence (page 3, line 99).

  1. On page 3, line 112, when the authors say that “showed a significant hypomethylation difference” they should present that difference and how it was calculated.

Response:

We analyzed the differences in methylation level between CRC tumors and normal tissues by using a paired t test (p < 0.001). The results have been added on page 3, line 112.

  1. The authors should clarify the exact location of the CpG sites instead of only say “gene body”. It is helpful to identify the CpG sites in the figures instead of putting 1, 2, 3, 4, 5, 6, 7, 8, 9, 10.

Response:

Thank you for this valuable suggestion. We added the numbers in the heatmap to improve data presentation. The CpG sites of the heatmap in this study were located exactly at the promoter and gene body. Sentences related to the CpG sites have been added on page 7 and to the legend of Figure 2: “Five CpG sites in promoter regions −1133, −525, −413, −122, and −94 are designated 1, 2, 3, 4, and 5, respectively. The CpG sites in gene body regions +12535, +927, +11421, +1331, and +2511 are designated 6, 7, 8, 9, and 10, respectively.”

  1. In Figure 2B only appear 19 patients, why? In the legend they say 20, this must be Again, in the text the authors only refer percentages, they should present statistical analysis.

Response:

We have replaced the data of a representative patient with a boxplot figure of the SMAD3 methylation level from 548 paired tissues and nine polyps. The results indicated significant differences between normal tissue with tumor and polyps with tumor (p ≤ 0.001, paired t test and nonparametric analysis). The bar graphs of methylation levels in each Taiwanese CRC patient have been presented in Supplementary Figure S1.

  1. Then in the QMSP, the authors only used 10 samples in 200 μL and 15 samples in 1 mL plasma, why?

Response:

Because of the limited number of available samples, the total plasma sample size was 50 individual specimens (30 participants for the 200 μL plasma analysis and 20 participants for the 1 mL plasma analysis). In this revision, we performed another eight ccfDNA extraction and methylation assays (1 mL of plasma). The sample sizes were increased to 28 for the 1 mL plasma analysis. The total sample size was 58 participants. This has been added to the Discussion section as a study limitation (page 14, lines 329–332), and relevant data have been added to the Results section (page 6, point 2.4).

  1. On page 5, line 139-140, the phrase should be re-phrased

Response:

The sentence has been modified to “We inferred that SMAD3 can be a predictive biomarker of CRC in Asian and Western populations” (page 5, lines 139–140).

  1. On page 6, the text under the point 2.4 need to be better explained. For example, what the authors mean by the part extracted using automatic beads in 1 mL of plasma?

Response:

We apologize for the unclear descriptions in this paragraph. In the QMSP analysis, we first tried to test the methylated level by using 200 μL of plasma. However, we found that the sensitivity (86.6%, 13/15) was low. To improve the detection sensitivity, ccfDNA extraction and methylation assays were performed in 1 mL of plasma from 28 different participants. The relevant sentence has been revised for clarity (page 6, lines 149–154).

  1. The text only refers to Figure 4 A and 4B, what about the results from figure 4C and 4D?

Response:

We apologize for the error. The text has been modified accordingly (page 6, lines 153).

  1. There are some symbols used in the figure 4A and 4C that are not explained in the figure legend.

Response:

A description of the symbols has been added to the legend of Figures 4A and 4C.

  1. Again, statistical analysis should be performed and presented, what the 20% in healthy means?

Response:

The phrase “20% in healthy” means that 2 of 10 (20%) healthy participants were found to have decreased methylation. The text under point 2.4 has been corrected to “When 1 mL of plasma was used, decreased methylation was detected in 7/10 (70%) patients with CRC but only 2/10 (20%) healthy controls (p = 0.038) (Figure 4C, 4D). The positive predictive value (PPV) of 1 mL of plasma was 64.7%, 11/17” (page 6, lines 149–154).

  1. The results in the text pag 7 lines 156 to 158 need to be modified. Is there a two-fold decrease or increase in SMAD3 expression in CRC tissues versus normal tissues?

Response:

We apologize for the typo. The word has been modified to “decreased.”

  1. When the authors say insignificant difference was observed, they should present the values and how they were obtained

Response:

The p values and statistical analysis results have been added to all the statistical process. (lines 176, 178, and 232).

  1. The authors when they refer to the results of mRNA expression of SMAD3 in the polyps samples they say 3/9 showed a low mRNA, is this which ones? because in the figure it seems that there´s low expression in polyps 2, 3, 4 and 5. Please explain. Also the statistical analysis is needed.

Response:

We have modified Figure 5A to form a boxplot. The expression levels in normal, tumor, and polyp tissues can now be clearly seen in this figure (page 8).

  1. The authors say “the trend of low expression…” did the authors done any statistical analysis to show if there is a difference between the expression levels in normal vs tumoral tissues using the TCGA data? If so, the authors should mention it.

Response:

The statistical data have been added to Figure 5B and the Results section. In 41 paired TCGA datasets, a significant difference was found between normal and tumor tissues in terms of expression levels (p ≤ 0.001, paired t test; page 7, point 2.5, lines 173–174).

  1. In the figure 5, the authors should think about a way to present the data using all the patients and not just show part. Maybe if they calculate the difference between the expression for each paired sample and then use this to statistically analysed in all the samples if there is a difference in fold change.

Response:

As suggested, the data of all patients have been presented in Figure 5 by using boxplots. Additionally, we present the data of all patients using a bar graph (Supplementary Figure S2).

  1. When the authors say: “ we found that hypermethylation of five CpG sites that are located at promoter regions cause low mRNA expression” the authors should also mention the two CpG sites in the gene body region that also give a correlation, again the location and identification of each probe would be positive for understanding better the results.

Response:

The following description has been added to Results section: “we found two CpG sites located in the gene body, namely cg07890839 (R = −0.350, p = 0.031) and cg03947447 (R = −0.440, p = 0.006), which also led to low expression” (page 7, point 2.5, lines 174–177).

  1. In the figure 3 the different colors should be identified in the legend

Response:

We apologize for leaving this out and have added color descriptions to the legend of Figure 3.

  1. In the figure 16: “19 normal controls and 19 paired patients with colorectal cancer” please explain better, which ones are the 19 normal controls?? Are they the paired of the 19 CRC samples or are those other samples?

Response:

The term “normal controls” here indicates the pairs of the 19 CRC samples; we have revised this term to “paired adjacent normal tissues” (page 9, line 194).

  1. On page 9, line 187-188, please rephrase: “SMAD3 can be detected not only in the late stage but also in the early stage” and “SMAD3 widely exists”. Please explain referring to the data obtained.

Response:

A comparison of SMAD3 methylation with clinical characteristics revealed that SMAD3 hypomethylation could be found in most of the parameters in both Taiwanese and TCGA datasets. The description “SMAD3 can be detected not only in the late stage but also in the early stage” has been clarified to “SMAD3 hypomethylation can be detected in both early and late stages of CRC (Tables 1 and S1)” (page 9, point 2.6, lines 198–199). Moreover, “SMAD3 widely exists” has been modified to “SMAD3 hypomethylation is commonly observed” (page 10, point 2.6, line 199).

  1. On page 9, line 189: “SMAD3 is a potential biomarker for early prediction of CRC” must be better explained, how the authors reach this hypothesis?

Response:

The alteration of SMAD3 mRNA can be observed in all stages of colorectal cancer. The results of methylation imply that SMAD3 can play a vital role in detecting CRC. However, the technical difficulties in assessing SMAD3 mRNA precludes its widespread application as a current potential biomarker.

  1. In the Table 1 (and table S1) only some characteristics from the tables are described in the text, while others are not such as: vascular invasion, location, MSI…

Response:

We apologize for the incomplete description. The sentence “Irrespective of age, sex, tumor type, tumor size, lymph node metastasis, distant metastasis, or differentiation grade, SMAD3 widely exists and is similar in several characteristics (Table 1 and Table S1)” has been corrected to “SMAD3 hypomethylation is commonly observed in several clinical parameters, such as age, ethnicity, sex, tumor type, tumor stage, tumor size, regional lymph node metastasis, distant metastasis, differentiation grade, vascular invasion, location, MSI, and KRAS mutation. A comparison of Taiwanese and TCGA datasets revealed similar results (Table 1 and Table S1)” (page 9, point 2.6, lines 199-202).

  1. The authors have checked methylation levels of SMAD3 in other cancers. Where did these samples came from? Some of these cancers, the sample size is too low to speculate about the methylation levels. For instance, there are only 2 samples from gastric cancer, 7 from rectal cancer and 12 from liver cancer.

Response:

All cancer samples were obtained from the TMU biobank. The limited number of available samples of other cancers is a study limitation. To verify the utility of SMAD3 in different cancers, we aim to use a larger sample size in further research.

  1. On page 13, line 307, what the authors mean by invasive investigation?

Response:

We apologize for the typo. The word “invasive” has been deleted from the sentence “This may be used for confirming the precancerous condition before further invasive investigation” in the Discussion section (page 13, line 288).

  1. On page 14, line 320, what the authors mean by “this phenomenon confirms that hypermethylation in promoter sites repress gene expression”?

Response:

Figure 6 indicates that promoter-site hypermethylation represses gene expression. The text “this phenomenon confirms that hypermethylation in promoter sites repress gene expression” has been corrected to “We propose that promoter-site hypermethylation represses gene expression” in the Discussion section (page 14, lines 295–296).

  1. The authors mention that using 200 μL of plasma the healthy participants showed false-positive results, it was not clear if this problem is no longer a problem when they used 1mL of plasma

Response:

The PPV of using 200 μL of plasma and 1 mL of plasma is 59% and 64.7%, respectively. The results suggest that increasing the volume can raise the PPV. However, the limited number of available samples is a study limitation. In a future study, we aim to increase the sample size and verify the utility of SMAD3 hypomethylation.

  1. Table 2 is presented in the discussion section, what is the reason for this? Some tumors have really low number, did the authors done statistical analysis using these data?

Response:

Table 2 has been moved to the Results section (pages 11–12).

  1. Line 353: “mRNA expression level of SMAD3 relative to GAPDH was two-fold higher” please explain this sentence better.

Response:

As suggested, we have modified the description “mRNA expression level of SMAD3 relative to GAPDH was two-fold higher” to “SMAD3 mRNA expression level was considered high if that of GAPDH was two-fold higher in colorectal tumor tissue than in normal colorectal tissue” for clarity (page 16, point 4.3, lines 379–381).

  1. There are various places which require grammatical correction. I have highlighted a few of these.
  2. a) 306 invasive check, replace this term
  3. b) the authors use the term “control group”, “patients”, “cases”, “cohort”, “specimens” these should be uniformized.

Response:

  1. a) The sentence “Detection of unmethylated SMAD3 ccfDNA in plasma can be used to predict CRC precancerous condition before invasive check” has been modified to “Further research will aim to increase the sample size to identify whether unmethylated SMAD3 ccfDNA in plasma can predict a CRC precancerous condition before colonoscopy and biopsy” (page 15, lines 358–360).
  2. b) The term “control group” is used only to describe the controls in the experiments, “cases” refers to people with a new diagnosis of cancer, and “specimens” specifically refers to the tissue or plasma samples from patients. The remaining words have been uniformized to “patients.”